# Leap-Of-Thought: Teaching Pre-Trained Models to Systematically Reason Over Implicit Knowledge

**Alon Talmor**[1,2]    **Oyvind Tafjord**[1]    **Peter Clark**[1]    **Yoav Goldberg**[1,3]    **Jonathan Berant**[1,2]

[1]The Allen Institute for AI
[2]Tel-Aviv University,  [3]Bar-Ilan University
{alont,oyvindt,peterc,yoavg,jonathan}@allenai.org

## Abstract

To what extent can a neural network systematically reason over symbolic facts? Evidence suggests that large pre-trained language models (LMs) acquire some reasoning capacity, but this ability is difficult to control. Recently, it has been shown that Transformer-based models succeed in consistent reasoning over explicit symbolic facts, under a "closed-world" assumption. However, in an open-domain setup, it is desirable to tap into the vast reservoir of implicit knowledge already encoded in the parameters of pre-trained LMs. In this work, we provide a first demonstration that LMs can be trained to reliably perform systematic reasoning combining *both* implicit, pre-trained knowledge and explicit natural language statements. To do this, we describe a procedure for automatically generating datasets that teach a model new reasoning skills, and demonstrate that models learn to effectively perform inference which involves implicit taxonomic and world knowledge, chaining and counting. Finally, we show that "teaching" the models to reason generalizes beyond the training distribution: they successfully compose the usage of multiple reasoning skills in single examples. Our work paves a path towards open-domain systems that constantly improve by interacting with users who can instantly correct a model by adding simple natural language statements.

## 1  Introduction

A longstanding goal of artificial intelligence is to develop systems that continuously accumulate knowledge by consuming facts and rules about the world and reasoning over them [1, 2, 3]. Recently [4], it has been shown that Transformers [5] are an effective architecture for this goal, as they can be trained to reason over knowledge expressed as natural language statements.

However, modern neural networks for language do not store knowledge symbolically. Instead, substantial amounts of knowledge are encoded in their parameters by pre-training on large corpora with a language modeling (LM) objective [6, 7, 8, 9]. Moreover, although LM-based models do exhibit certain reasoning abilities [10, 11], these abilities are not systematic and are difficult to control. Thus, for real-world open-domain applications, it is imperative that models reason consistently over both *explicit* input statements and *implicit* knowledge that is already encoded by the network.

In this work, we develop models that reason over implicit knowledge and explicit natural language statements in a systematic manner. Consider the example in Figure 1. The model needs to determine whether *"A whale has a belly button"*, but answers incorrectly since this particular knowledge nugget is unknown to the model. However, if a user tells the model explicitly that *"A mammal has a belly button"*, the model can combine this statement with the its implicit knowledge that *"A whale is a mammal"*, and make the right inference on the spot. Thus, we want the model to systematically handle such cases that combine implicit knowledge with input natural language statements.

We train our models by automatically generating examples that illustrate the expected types of inference. Because the knowledge of pre-trained models comes from real world text, we test this knowledge by generating examples using true facts and rules from multiple information sources.

We focus on two types of high-level reasoning: (a) inference that combines implicit taxonomic knowledge (hypernymy, meronymy, etc.) with explicit natural language rules, and (b) inference that requires counting over both implicit and explicit facts, and checking whether a certain count was reached. In both cases, we observe

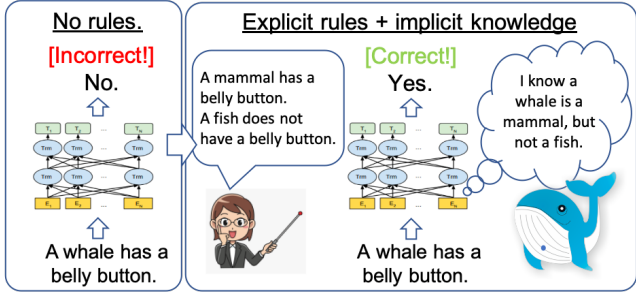

Figure 1: The model is wrong when asked whether *"A whale has a belly button"*. However, if a user tells the model the explicit rule *"A mammal has a belly button"*, the model combines this on-the-fly with its implicit knowledge that *"A whale is a mammal"*, and arrives at the right conclusion (without re-training).

that models can be trained to reason over implicit and explicit knowledge. Importantly, model performance can be explained by its prior knowledge: inference is successful when the necessary knowledge exists in the model, and fails when it is missing.

Last, we show that training the models to perform inference generalizes *beyond* their training distribution. Specifically, we endow a pre-trained LM with multiple inference capabilities independently, and show that it can handle examples that require composing multiple inference types, even when these do not appear at training time. Thus, one can gradually improve the inference capabilities of a model and expect generalization.

Our work paves a path towards systems that constantly improve by interacting with users: when a user spots an error, they can fix that error by providing a single statement in natural language that will allow the model to apply its acquired inference skills and reach the right conclusion. If the system can successfully retrieve this statement in future interactions, it will fix not only the current mistake, but also future ones. This can be viewed as a form of "one-shot learning" that improves the model on-the-fly without further training, unlike most current work that relies on data collection and re-training for fixing model errors. All our code and data is publicly available at `http://github.com/alontalmor/LeapOfThought`.

## 2 A Motivating Example

We begin by demonstrating that combining reasoning over explicit input with reasoning over implicit LM knowledge can "emerge" by training a Transformer on a dataset that requires each skill individually. We fine-tune ROBERTA [8], on binary (yes/no) question answering tasks from two datasets (using standard multi-task training): (a) 50K examples from TWENTY QUESTIONS (20Q),[1] a question answering (QA) dataset which includes questions such as *"Does an aircraft fly?"* (`true`) and *"Do everyone have an alarm?"* (`false`). This teaches the model to retrieve real world facts from its internal implicit knowledge; and (b) 100K examples from the RULETAKER [4] reasoning dataset, teaching the model to reason over a set of assertions explicitly provided as natural language statements.

We evaluate this model on a task that requires combining implicit knowledge about the sizes of animals (known to exist in ROBERTA [11]) and an animal taxonomy, with explicit reasoning over natural language statements. Figure 2 illustrates the setup. The model needs to determine if a

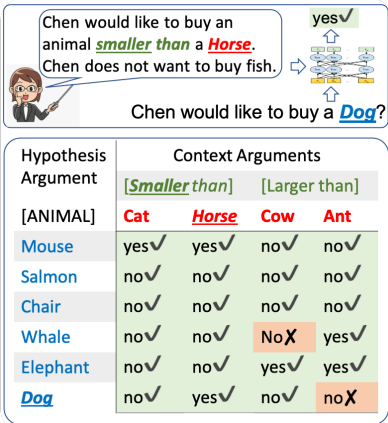

| Hypothesis Argument | Context Arguments | | | |
|---|---|---|---|---|
| | [*Smaller* than] | | [Larger than] | |
| [ANIMAL] | Cat | *Horse* | Cow | Ant |
| Mouse | yes✓ | yes✓ | no✓ | no✓ |
| Salmon | no✓ | no✓ | no✓ | no✓ |
| Chair | no✓ | no✓ | no✓ | no✓ |
| Whale | no✓ | no✓ | No✗ | yes✓ |
| Elephant | no✓ | no✓ | yes✓ | yes✓ |
| *Dog* | no✓ | yes✓ | no✓ | no✗ |

Figure 2: Our motivating task. The top box shows a single example. In the bottom we systematically replace arguments matching the color of the bold underlined words in the top example. The text in the table corresponds to model predictions, and the color and ✓ indicate a correct prediction.

[1]`https://github.com/allenai/twentyquestions`

hypothesis of the form *"Chen would like to buy a [ANIMAL]"* is true, where the slot *ANIMAL* is replaced with some object. The model also observes explicit knowledge that specifies the desired size of the animal and that it must not be a fish. Over 24 animal pairs, the model obtains 91.6% accuracy, successfully combining implicit knowledge with the given statements in ways that were not observed during fine-tuning. Variations of this task work equally well.

This experiment shows that although training examples did not require reasoning over explicit statements *and* implicit knowledge, the model learned to effectively do so. While exciting, the procedure is still an "alchemy". Can we make it more systematic and more controlled, and get a better handle on its capabilities and limitations? The rest of the paper explores these questions.

## 3   Method

We describe a general procedure for creating models that can perform inference over explicit and implicit knowledge. The process includes automatic data generation from existing sources, followed by standard supervised training. §4.1 and §4.3 show two instantiations of this procedure.

**Definitions**   Our goal is to endow a model with the ability to perform certain *inference types*. An *inference type* describes how two or more natural language statements combine logically. For example, we define the HYPERNYMY inference type with the assertion 'if A is a type of B, and B has property P, then A has property P'. Similarly, we will later use APPROX-IMATE COUNTING, and MERONYMY as additional inference types. To this end, we automatically generate training examples, where each example includes (a) a *hypothesis*, i.e., a textual statement that is either true or false (*"A whale has a belly button"*), and (b) *explicit knowledge*, which is a list of textual statements. Statements can either be *facts*, that is, describe a property of a particular entity (*"John Lennon is a member of The Beatles"*), or *rules*, that is, describe a property of some class (*"Mammals have belly buttons"*). The explicit knowledge is constructed such that the truth value of the hypothesis cannot be inferred from the explicit knowledge alone, but also requires some knowledge to be encoded by the LM a-priori. For example, the explicit knowledge might not include the rule *"A whale is a mammal"*, necessary for deducing that they have belly buttons (Figure 1).

**Data generation**   Our primary motivation is to develop models that work in an open domain environment with real world facts. Thus, we automatically generate data by sampling from existing knowledge sources: CONCEPTNET [12], WORDNET [13] and WIKIDATA [14]. We sample (`subject`, `predicate`, `object`) triples that are known to be either `true` or `false`. Pseudo-language statements are then generated using manually constructed templates for each `predicate`.[2] For example, for true statements such as (`chicken`, `has`, `feathers`) we generate *"A chicken has feathers."*; for false statements, we generate *"A chicken does not have horns"*. In all examples, the order of the statements in the explicit knowledge is random and the number of true and false hypotheses is equal.

**Training**   Once examples are generated, we fine-tune a pre-trained LM, specifically ROBERTA-LARGE [8], on this dataset. The inputs and outputs are modeled in the standard manner [7]: The input is given as a list of tokens '`[CLS] explicit knowledge [SEP] hypothesis [SEP]`', and the contextualized representation of the `[CLS]` token is linearly projected down to two logits, and passed through a softmax layer to obtain the probabilities that the hypothesis is true or false. We train by minimizing the binary cross-entropy loss, and evaluate models using accuracy.

In addition, To investigate the importance of pre-trained contextualized representations, we use ESIM [15] over non-contextualized GLOVE [16] representations as a baseline architecture, as it is known to provide a strong model when the input is a pair of text fragments [15]. We adapt the architecture to the mutli-choice setup using the procedure proposed by [17], with two choices for 'yes' and 'no'. Last, motivated by the results in §2, we evaluate whether the RULET.+20Q model, described in §2, can utilize implicit knowledge even without being directly trained for this goal.

# 4 Experiments

We instantiate our approach over two inference types. First, correcly applying implicit taxonomic knowledge (§4.1). Second, counting over explicit and implicit facts (§4.3). Additionally, we analyze our results and show that model success can be reliably predicted by probing the background knowledge encoded in the pre-trained LM (§4.2).

## 4.1 Implicit Knowledge of Taxonomic Relations

Pre-trained LMs have been shown to capture substantial amounts of taxonomic knowledge such as hypernymy (`A is a type of B`) and meronymy (`A is part of B`) [10]. To test whether this knowledge can be leveraged for reasoning, we create examples that can be solved only by integrating this implicit knowledge.

**Data Construction** Figure 3 illustrates the main components of an example. We describe the generation process, based on triples from CONCEPTNET and WORDNET.

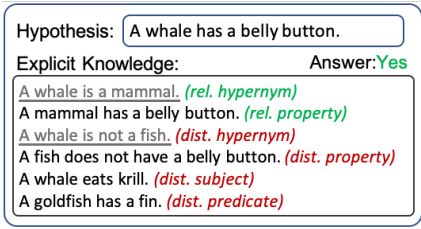

Figure 3: Outline of a taxonomy example. The purpose of each relevant and distractor rule is in parenthesis. Underlined hypernym rules are removed in IMPLICIT REASONING.

To generate a positive example (`true` label), we sample a `relevant hypernym` rule, e.g., ((`whale, is a, mammal`), `true`). We then find a `relevant property` of the hypernym object, e.g., ((`mammal, has a, belly button`), `true`). We apply the hypernymy inference type (downward monotonicity): (`if A is a B and B has property C, then A has property C`) to deduce the logical conclusion which will become the hypothesis, e.g., ((`whale, has a, belly button`), `true`).

To add distractors, we create similar but irrelevant statements by: (a) randomly replacing the `subject` of the `relevant property` to create a `distractor property`, e.g., ((`fish, has a, belly button`), `false`). (b) Create a `distractor hypernym` by combining the subject of the hypothesis and the subject of the `relevant property`: e.g., ((`whale, is a, fish`), `false`), (c) add a `distractor subject` by randomly sampling a different rule about the hypothesis `subject`, and (d) add a `distractor predicate` by randomly sampling a different rule with the hypothesis predicate. Thus, the explicit knowledge contains the relevant hypernym and property, and four distractors.

Negative examples are created from positive examples by using the `distractor property` (*"a fish does not have a belly button"*). We sample a hypernym rule, such that the object matches the subject of the `distractor property` e.g. ((`salmon, is a, fish`), `true`). We then apply the hypernymy inference type to obtain the false hypothesis ((`salmon, has a, belly button`), `false`). In the negative example the roles of the relevant hypernym and property, and distractor hypernym and property are reversed. The distractor subject and predicate are sampled independently.

We generate 30,906 training examples using this procedure. We create development and test sets, 1,289 examples each, where the subjects and objects are *disjoint* from the ones in the training set. In 50% of the training examples, we remove the relevant and distractor hypernyms to teach the model to use implicit knowledge. We find that without this step, the model always predicts false if a necessary hypernym is missing, performing well only on the EXPLICIT REASONING setup. In 20% of the training examples we remove all distractors.[3] In the test set, we remove some of the statements in the explicit knowledge, based on the experimental setting, as explained below. For example, we remove all hypernyms to test the use of implicit knowledge.

***Meronyms (zero-shot)*** To test whether the inference type learned by the model generalizes to other inference types, we used the same procedure to create 2,528 test examples using a MERONYMY inference type (meronym transitivity), i.e. '`if A has part B and B has part C, then A has part C`' or '`if a hand has a cell and a cell has an atom, then a hand has an atom.`'. There is no training set in this setup.

**Experiments**   We evaluate our model in three different setups:

HYPOTHESIS-ONLY: The model is given only the hypothesis without the explicit knowledge. This tests whether the model already knows the answer without the need for inference.

EXPLICIT REASONING: The model is given the hypothesis and explicit knowledge with hypernym rules, thus the model needs to perform inference over explicit natural language statements.

IMPLICIT REASONING: The model is given the hypothesis and explicit knowledge *without* hypernym rules, and is forced to choose the correct answer based on its implicit knowledge.

**Models**   We compare the following models: (a) ROBERTA fine-tuned on the generated dataset, (b) The RULET.+20Q model (see §3), and (c) ESIM trained on the generated dataset without pre-training (testing the value of pre-trained contextualized representations).

**Results**   We show the results in Table 1. On HYPOTHESIS-ONLY evaluation examples, ROBERTA achieves 65.2% accuracy, substantially higher than random (50%), suggesting it knows some of the answers even without the explicit knowledge (if we train only on HYPOTHESIS-ONLY examples, accuracy on HYPOTHESIS-ONLY test examples is slightly higher at 69.7%). EXPLICIT REASONING shows that when the relevant hypernym and property are both present in the explicit knowledge, ROBERTA achieves near perfect performance, easily solving 2-hop reasoning. Finally, the accuracy of IMPLICIT REASONING is 88.8%, substantially higher than HYPOTHESIS-ONLY. This suggests the model is able to correctly apply its implicit knowledge of hypernyms to select the relevant property rule, effectively performing 2-hop reasoning with one hop done internally.

We evaluate performance of RULET.+20Q to determine if combining implicit and explicit knowledge emerges even without direct training.[4] The model, although trained on a different distribution for explicit reasoning (the RULETAKER dataset), achieves 98.4% on EXPLICIT REASONING. Surprisingly, without being trained to perform implicit reasoning, accuracy improves from $65.4 \rightarrow 79.1$ when given the explicit knowledge without hypernym rules (but still lower than 88.8). ESIM, trained with

| Model $\rightarrow$ | ROBERTA | | | ESIM |
|---|---|---|---|---|
| Train-set $\rightarrow$ | Hyper. | Hyper. | RULET.+20Q | Hyper. |
| Test-set $\rightarrow$ | Hyper. | Mero. | Hyper. | Hyper. |
| HYPOTHESIS-ONLY | 65.2 | 70.8 | 65.4 | 61.3 |
| EXPLICIT REASONING | 99.7 | 99.4 | 98.4 | 79.8 |
| IMPLICIT REASONING | **88.8** | **86.9** | **79.1** | **76.1** |

Table 1: Test set results for reasoning over hypernymy and meronymy relations. The models learn to reason with implicit rules, significantly improving on the hypothesis-only baseline, some in zero-shot.

GLOVE pre-trained word embeddings, also shows a higher accuracy of 76.1% in IMPLICIT REASONING compared with 61.3% in HYPOTHESIS-ONLY. This suggests leveraging implicit knowledge is possible in models with non-contextualized representations. However, ESIM did not achieve perfect accuracy in EXPLICIT REASONING, reaching 79.8%. Interestingly, performance on meronymy is similar to hypernymy although no meronyms were provided during training, suggesting the model already has some knowledge of this reasoning skill from pre-training.

### 4.2   Analyzing systematicity

IMPLICIT REASONING achieves 88.8 accuracy, a high, but not perfect, result. A key question is whether the performance in IMPLICIT REASONING can be explained in terms of the implicit knowledge of the LM. To estimate a model's implicit knowledge "beliefs", we convert the required implicit knowledge (relevant and distractor hypernyms) to hypotheses without any explicit knowledge rules (i.e. only *"A whale is a mammal"* hypothesis), and run the model to check if prediction is correct. We hypothesize that model performance should be high for cases where the implicit beliefs are accurate.

Table 2 shows how scores depend on whether all the model's implicit beliefs are correct. For the hypernymy examples, the model reaches 99.7 accuracy when the implicit beliefs are both correct, while it is much lower otherwise. Using the knowledge of which beliefs are incorrect, we can intervene and explicitly add the correct relevant and distractor hypernyms for the small number of incorrect beliefs (17% in this case). With intervention, the score goes up from $88.8 \rightarrow 98.3$, a significant error reduction.

A potential bias in the setup from §4.1, is that the model also has an initial belief about the correctness of the hypothesis itself, regardless of its belief of the hypernym rules. To neutralize this effect we create a controlled dataset IMAGINARY, where entities have imaginary properties, such as *"group-1"* in *"A Toucan is a group-1"*, for which the model does not have any prior knowledge. The rest of the dataset construction is similar to §4.1. Thus, the performance of HYPOTHESIS-ONLY in this setup is 50% by construction. As before there is one relevant implicit rule (*"Toucan often has part wing."*), but now there can be up to 5 distractor implicit rules, as well as a mix of rule conditions (hypernyms, meronyms and size

| Implicit beliefs | Hypernymy | IMAGINARY |
|---|---|---|
| All correct | 99.7 | 95.0 |
| Some incorrect | 70.0 | 66.3 |
| All incorrect | 13.0 | 7.1 |
| Overall | 88.8 | 76.9 |
| Overall (after intervention) | 98.3 | 94.1 |
| Fraction beliefs corrected | 0.17 | 0.20 |

Table 2: When the model's implicit beliefs needed for a hypothesis are all correct, scores are high. If we intervene for the small number of incorrect beliefs, adding their rules to the explicit knowledge, the overall scores increase accordingly.

comparisons). In Table 2, we observe a similar consistency in results when predicting the outcome for the IMAGINARY-set, where accuracy is $95.0$ when all beliefs are supporting the correct answer, $66.3$ when there are conflicting beliefs, and $7.1$ when all beliefs are in support of the wrong answer. Intervening here improves overall performance from $76.9 \rightarrow 94.1$. More details on IMAGINARY are available in the supp. material. To conclude, we observe that model reasoning can be explained by its prior knowledge, and that one can correct it by intervening and correcting false beliefs, suggesting that the model is using implicit knowledge in a systematic fashion.

## 4.3 Counting over Implicit Facts

Our next experiment focuses on whether our models can *simulate* counting over explicit and implicit facts. While LMs are known to hold taxonomic knowledge [10], there is evidence that skills such as counting are not acquired during pre-training [11]. Here, we train a model for simulating counting, and check whether it can count over both text and its prior knowledge.

**Data Construction** To collect 'facts that require counting' (*member facts*), we use WIKIDATA relations such as *"member of band"*, *"capital of country"*, and *"child of person"*. *Quantity facts* provide the total count

> **Hypothesis:** Mick Jagger is a member of the Beatles.
>
> **Explicit Knowledge:**                **Answer:** No
> Ringo Starr is a member of The Beatles. *(member fact)*
> John Lennon is a member of The Beatles. *(member fact)*
> The Beatles has four members. *(quantity fact)*
> Jeff Bezos is CEO of Amazon. *(dist. member fact)*
> Amazon has one CEO. *(dist. quantity fact)*
> The Carpenters has two members. *(dist. quantity fact)*
> Karen Carpenter is a member The Carpenters. *(dist. member fact)*

Figure 4: Fact counting example outline. The purpose of each fact is in parenthesis.

of facts for a specific entity, e.g., *"The Beatles has **four** members"*. Figure 4 outlines a counting example. For each entity (*"Beatles"*), there are 1-4 member facts (*"members"*).

We create training data by constructing 10,852 *counting sets*, that is, sets that include a single quantity fact (*"The Beatles has four members."*) stating the total number of member facts and $K$ member facts, (*"Ringo Starr is a member of The Beatles"*), etc. For each counting set, we create exactly $2K - 1$ examples: For all integers $k \in [0, K - 1]$, we create an example by adding $k$ member facts and the quantity fact to the explicit knowledge, and generating one positive hypothesis and one negative hypothesis. The positive hypothesis is a true member fact not in the explicit knowledge, and the negative hypothesis is generated by randomly sampling a `subject` that appears with the relevant `predicate`, e.g. ((Mick Jagger, is member of, The Beatles), false). When $k = K$, we generate only a false hypothesis.

Distractors are added to the explicit knowledge by adding one quantity fact and one member fact that share a predicate with the hypothesis, but have a different subject (*"The Carpenters"*), and one random member fact and quantity fact (*"Jeff Bezos is CEO of Amazon"*). We find that distractors are important in the sense that if none are added, the model learns to count sentences, and ignores their content, failing to pay attention to the actual subjects of the member facts.

Overall 38,700/3,005/3,005 training/development/test examples were created. In 25% of the examples $k = K$ and thus no implicit knowledge is needed. If the model can count the relevant member facts and verify the count reaches the number specified in the relevant quantity fact, then it can deduce that the answer is `false`. In all other examples, solving the task requires combining the member facts that appear in the explicit knowledge with implicit member facts that the model knows.

**Experiments** We evaluate our approach using ROBERTA and ESIM (the performance of RULET.+20Q is low since it cannot count without training).

HYPOTHESIS-ONLY: The model is given only the hypothesis (similar to §4.1).

COUNTING: Examples are sampled from the same distribution as the training set. In 25% of the examples the model can use explicit counting to predict `false` ($k = K$). In the rest, it must combine prior knowledge of member facts with member facts that are in the explicit knowledge.

**Results** Table 3 shows the accuracy for all models. We distinguish between the case where the total number of member facts has been reached ($k = K$), where the answer is `false`, and the rest of the examples. ROBERTA achieves a near perfect result of 99.7% when $k = K$, illustrating it can be trained to count the relevant member facts.[5]

When the number of member facts is in $1, \ldots, K - 1$, accuracy improves by 9 points given the explicit knowledge ($64.1 \to 73$), implying that the model learned to combine implicit knowledge of member facts with the explicit knowledge. Both RULET.+20Q (not trained on the counting dataset) and ESIM, trained on the generated data, predicted `false` to almost all examples, indicating they are not well suited for this task. To further validate that the model is indeed counting, we dropped the relevant quantity fact from all test examples. The accuracy drops from $73 \to 64.4$ (Table 3, counting $(1, K - 1)$), which is similar to hypothesis-only $64.1$, suggesting that the model is using the quantity fact to know how many member facts should be counted.

| Experimental setup | Subset | ROBERTA |
|---|---|---|
| HYPOTHESIS-ONLY | $(1, K - 1)$ | 64.1 |
| COUNTING | $(1, K - 1)$ | 73 |
| | $K$ | 99.7 |

Table 3: Test set performance for counting. We show performance on two subsets of the test set: $(1, K - 1)$: where the number of member facts is in $1, \ldots, K - 1$, and $K$ where the number of member facts is exactly $K$.

Performance shows that the model uses the explicit knowledge to improve prediction. However, because the required implicit knowledge involves long-tail facts that might be missing from the LM, learning to use this knowledge can be difficult. We perform a more fine-grained analysis to decouple prior knowledge from the counting task at hand.

**Analysis** Our analysis setup is the following. We take all member facts, and obtain a *fact probability*, which is the output of our model when given the member fact as a hypothesis without any explicit knowledge. Then, for each counting set we create $2K - 1$ examples as before, but for each $k \in [1 \ldots K - 1]$ we take $k$ facts not *randomly*, but according to one of two orders: (a) ascending fact probability, and (b) descending fact probability. We hypothesize that if member facts are sorted in ascending order, performance should be higher, because the model is given explicit evidence for facts it is less confident about.

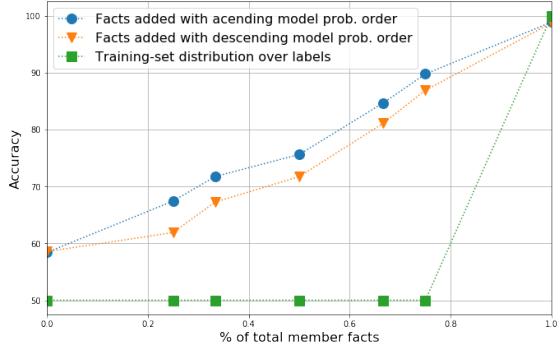

Figure 5: An analysis of ROBERTA accuracy (y-axis) vs $c = \frac{k}{K}$ (x-axis).

Figure 5 shows the results. The x-axis represents $c = \frac{k}{K}$, that is, the fraction of member facts given in the explicit knowledge. When $c = 0$ or $c = 1$, ascending and descending orders are equivalent, since either no facts or all facts are given. Green squares show the distribution over labels for each value of $c$, illustrating that guessing based on label leads to 50% accuracy, except when $c = 1$, in which all examples are `false`. Blue circles display accuracy when member facts are added in ascending order, and orange triangles in descending order.

When $c = 0$, model accuracy is slightly lower than 60%, showing that ROBERTA has non-trivial knowledge of the relevant background facts. When $c = 1$ performance is perfect since our models count the correct facts and reach $K$. For both ascending and descending orders, the accuracy of

ROBERTA monotonically increases with the number of member facts, although distribution over labels is constant. This suggests that explicit member facts help the pre-trained LM leverage its prior knowledge. When $c = 0.75$ both models hover around an impressive 90% accuracy. We confirm our hypothesis that adding facts in ascending order substantially improves performance: when $c = 0.25$ performance improves by 5.5 points, and when $c = 0.75$ it improves by 2.7. This shows that adding member facts which the model does not "know" improves performance more than a fact it is already confident about.

## 4.4 Generalizing to New Skill Combinations

In §4.1 and §4.3, we isolated a single inference type in each experiment. However, our goal is to have a model that can seamlessly combine these skills. To test this, we train models on various subsets from RULETAKER, 20Q (described in §2), COUNTING (§4.3) and HYPERNYMS (§4.1). We then test whether these models can handle examples that require multiple skills simultaneously. Training is done by randomly mixing the datasets and performing standard multi-task training.

Ultimately, we want users to correct models on-the-fly by mixing natural language statements that demand different reasoning skills. To emulate this process, we created MULTI-SKILL-SET, composed of 185 hand-crafted hypotheses and explicit knowledge,[6] labeled with the set of skills needed for each example. Figure 6 shows one example, requiring age/year comparison + hypernymy skills. We specifically chose hypotheses ROBERTA answers incorrectly in HYPOTHESIS-ONLY.

**Results** Table 4 shows results on MULTI-SKILL-SET. The rows show the subset of datasets the model was trained on and the columns show the accuracy for examples that require a particular skill. A model trained on all 4 datasets obtains an accuracy of 85.4. All models perform poorly given only the hypothesis, with an average accuracy of 40.2. (Less than random, due to the adverserial manner in which we chose examples for MULTI-SKILL-SET). Models trained without COUNTING show low accuracy on the counting skill subset, suggesting that this skill was not acquired at pre-training time. Interestingly, models combining HYPERNYMS or COUNTING with RULETAKER and 20Q, display higher accuracy than models trained only on one of these datasets. This suggests that models are able to successfully combine multiple reasoning skills, some newly acquired. Models show high accuracy on skills that they are not explicitly trained for, but have been shown to be acquired during pre-training, such as size comparison [11], and meronyms [18].

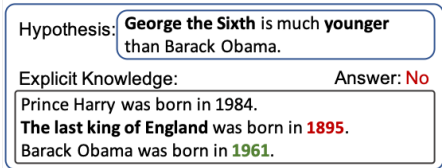

Figure 6: MULTI-SKILL-SET example, requiring age/year + hypernymy skills.

The right side of Table 4, presents the accuracy on examples requiring at least two skills. Strikingly, the model is able to accurately compose multiple skills, reaching a high accuracy of 88 on examples combining implicit hypernyms and counting. Size, age, and year comparison examples are not available in our training sets, nevertheless, models achieve an impressive score of 81.8 and 73.9 on examples combining implicit hypernyms with size and age/year, respectively.

| test questions, skills → <br> training data ↓ | overall | hypothesis only | hypernyms | counting | sizes | age/year | hypernyms + counting | hypernyms + sizes | hypernyms + age/year |
|---|---|---|---|---|---|---|---|---|---|
| HYPERNYMS | 62.7 | 50.8 | 64.7 | 34.0 | 69.2 | 46.7 | 28.0 | 72.7 | 43.5 |
| COUNTING | 66.5 | **54.6** | 65.5 | 72.3 | 46.2 | 63.3 | 60.0 | 45.5 | 69.6 |
| RULET.+20Q | 69.2 | 34.6 | 76.5 | 55.3 | 73.1 | 60.0 | 68.0 | 68.2 | 60.9 |
| HYPERNYMS+RULET.+20Q | 73.5 | 31.9 | 74.8 | 44.7 | 61.5 | 76.7 | 48.0 | 63.6 | **73.9** |
| COUNTING+RULET.+20Q | 70.8 | 34.6 | 68.1 | **83.0** | 65.4 | 56.7 | **88.0** | 59.1 | 52.2 |
| ALL COMBINED | **85.4** | 34.6 | **84.0** | **83.0** | **84.6** | **80.0** | 84.0 | **81.8** | **73.9** |

Table 4: Accuracy for MULTI-SKILL-SET. Rows show training set composition for ROBERTA. Overall accuracy, hypothesis-only accuracy, single-skill and multiple-skill breakdown are displayed in columns.

# 5   Related Work

We have demonstrated that LMs can be taught to *systematically* combine both pre-trained knowledge and explicit natural language statements. This predictability is important: If the model is behaving rationally, we can teach it. This distinguishes our work from other research in recognizing textual entailment (RTE), e.g., [19, 20, 21, 22], where models learn to score a hypothesis but in an opaque and somewhat unpredictable way [23, 24]. It similarly distinguishes our work from multi-hop QA, e.g., [25, 26, 27, 28, 29], where again model behavior is often opaque [30].

There have been numerous earlier demonstrations that neural systems can learn systematic behavior, including for semantic parsing [31], symbolic integration [32], mathematics [33], knowledge prediction [34, 35], and rule-based reasoning [4]. However, these are all largely self-contained tasks. We extend this to show how implicit knowledge can be directly harnessed in a systematic inference process.

Finally, although teaching a machine via general statements has long been a goal of AI [36], current neural methods typically require large numbers of examples to convey knowledge [37]. Our work shows how pre-trained networks can instead be taught on-the-fly using a few *general* statements, in a "one-shot" manner that exploits pre-trained knowledge, without requiring re-training of the model.

# 6   Discussion

In this work, we show that pre-trained LMs can be trained to consistently combine implicit knowledge encoded in their parameters with explicit rules and facts. Models are able to perform various types of reasoning in this setup including multi-hop reasoning, counting, number comparisons, and taxonomic knowledge. Moreover, we show that one can inject the ability to perform various types of inferences one at a time independently, and obtain generalization to cases that require combining these skills in a single example. Our work opens the door to models that learn through interaction with users. Users can teach the model facts and rules about the world through natural language statements, and the model will utilize this new information immediately, combining it with the knowledge encoded internally. Such an approach allows users to "teach the model" and correct its current and future errors without the need for data collection and re-training, in a one-shot manner.

## Broader Impact

Our work, if successful, paves a possible path towards models that learn in a one-shot manner by interacting with users. Users of Question Answering and other systems utilizing reasoning over natural language may benefit by constantly improving models that do not require re-training in an interactive manner. However, users teaching false rules and facts may lead to the spread of ungrounded and possibly "fake" information. Thus, the provided rules and facts must be constantly monitored and curated. Finally, users relying on the reasoning of such systems for mission critical tasks, such as medical advice, might be at risk from possible errors. At current level of accuracy of state-of-the-art models, this type of usage is not advised.

## Acknowledgments and Disclosure of Funding

We thank our colleagues at The Allen Institute of AI and Tel-Aviv University, especially Kyle Richardson, Nicholas Lourie, Ashish Sabharwal, Elad Segal, Mor Geva and Tomer Wolfson. This research was partially supported by The Israel Science Foundation grant 942/16, The Blavatnik Computer Science Research Fund and The Yandex Initiative for Machine Learning, and the European Union's Seventh Framework Programme (FP7) under grant agreement no. 802774-ERC-iEXTRACT and no. 802800-DELPHI.

## Footnotes

[2]The full list of predicates used is available in the supplementary material.

[3]The training set also includes 2,864 examples that contain only a hypothesis, half of which are true hypernym rules (*"a whale is a mammal"*) and half are false hypernym rules (*"a whale is a fish"*). This is useful for analyzing what prior knowledge a LM has, as we show in §4.2.

[4]Clark et al. [4] tested whether ROBERTA can perform reasoning in the few-shot setup, without further fine-tuning on additional auxiliary datasets, and found that models perform poorly (figure 4 in their paper).

[5]We also conduct an experiment where we provide $K$ member facts, but some of them are false (e.g., *"Mick Jagger is a member of the Beatles"*) and find that the model simply counts member facts, ignoring whether they are factually correct or not, still predicting `False` in all such cases.

[6]All examples and best model prediction are available in the supplementary material.

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
