[Supplementary Material]

# Teaching Pre-Trained Models to Systematically Reason Over Implicit Knowledge - Supplementary Material

## 3   Method

Pseudo-language statements are generated using manually constructed templates for each `predicate`. The predicates used include the CONCEPTNET relations (`IsA, Antonym, DistinctFrom, PartOf, CapableOf, Desires, NotDesires`). In addition, (`IsHypernymOf, IsMeronymOf`) relations were produced by combining WORDNET and CONCEPTNET, for the Hypernymy and Meronymy datasets.

For the COUNTING task, we use the following predicates for the member facts: (`super bowl winner, super bowl loser, band member, capital, director, release year, founder, headquarter, child, spouse,CEO`). For quantity facts, predicates include (`has 1,..,has 5`), effectively supporting up to a count of five member facts.

## 4   Experiments

### 4.1   Implicit Knowledge of Taxonomic Relations

**Experiments**   We evaluate our model in two additional setups:

STATEMENT-ONLY-NO-CONTEXT: We compare the effect of removing the explicit knowledge entirely verses the HYPOTHESIS-ONLY setup, in which only the relevant rules are removed.

STATEMENT-ONLY-LANGUAGE-SELECTIVITY: To insure the model is using the hypothesis subject, i.e. *"Whale"* in *"A whale has a belly button"*, we replace the subject of the statement with random words that carry no meaning such as *"foo, blah, ya, qux, aranglopa, foltopia, cakophon, baz ,garply"*. This experiment controls for weather the model can answer correctly solely based on the context.

**Results**   Table 1 compares the two new experiments introduced. STATEMENT-ONLY-NO-CONTEXT shows slightly higher results on average, suggesting that the distractors in the explicit knowledge partially mislead the model compared to a case where no explicit knowledge is used.

| Model $\rightarrow$ | ROBERTA | | | ESIM |
|---|---|---|---|---|
| Train-set $\rightarrow$ | Hypernymy | Hypernymy | RULET.+20Q | Hypernymy |
| Test-set $\rightarrow$ | Hypernymy | Meronymy | Hypernymy | Hypernymy |
| HYPOTHESIS-ONLY | 65.2 | 70.8 | 65.4 | 61.3 |
| STATEMENT-ONLY-NO-CONTEXT | 66.7 | 71.1 | 68.0 | 59.0 |
| STATEMENT-ONLY-LANGUAGE-SELECTIVITY | 55.6 | 55.5 | 57.7 | 54.9 |

Table 1: Test set results for reasoning over hypernymy and meronymy relations. The models learn to reason with implicit rules, significantly improving on the hypothesis-only baseline, some in zero-shot.

In the STATEMENT-ONLY-LANGUAGE-SELECTIVITY experiment, the model achieves a slightly higher than random accuracy of 55%, implying that the explicit knowledge does give away the

answer to a small number of examples but in the rest the full hypothesis is needed to arrive at the correct answer.

## 4.2 Analyzing systematicity

**Details on IMAGINARY dataset** In the "real world rules" experiments presented in Section 4.1, the model may also have a prior belief about the *answer* from pre-training, adding a potentially confounding element, e.g., is the model right because of reasoning or prior knowledge?

To factor this out, we perform another experiment using the IMAGINARY dataset (Figure 1) with neutral rule conclusions of the form *"X is a group-N"*. We generate rule sets where, for the entity in the hypothesis, only one condition is true (relevant fact) while 3-5 others are false (distractor facts). We divide the rules randomly into positive and negative rules. We mix three different relation types (hypernym, meronym, size comparison), and distractors are sampled to be somewhat adversarial (inverse relation, other relation, etc). For the IMPLICIT REASONING setting, we exclude all the facts from the context.

We train a ROBERTA model as in Section 4.1, on a mix of settings (EXPLICIT REASONING, IMPLICIT REASONING, andHYPOTHESIS-ONLY for the knowledge facts). The development and test sets use a disjoint hypernym tree of entities from the training set. As expected, this model scores near-random (52.3%) on the HYPOTHESIS-ONLY variant. It scores near-perfect on EXPLICIT REASONING (99.0%), while scoring 79.2% on the IMPLICIT REASONING variant.

Figure 1: Example from the IMAGINARY dataset. Underlined facts are removed in IMPLICIT REASONING.

**Hypothesis:** Toucan is a group-1.
**Explicit Knowledge:** **Answer:** Yes
Toucan often has part wing. *(rel. fact)*
Toucan is not a material. *(dist. fact)*
Toucan never has part carapace. *(dist. fact)*
Toucan is smaller than a horse. *(dist. fact)*
If something often has part wing, then it is a group-1. *(rel. rule)*
If something is a material, then it is not a group-1. *(dist. rule)*
If something often has part carapace, then it is not a group-1. *(dist. rule)*
If something bigger than a horse, then it is a group-1. *(dist. rule)*

## 4.4 Generalizing to New Skill Combinations

To further explore the systematicity of reasoning over multiple skills, we create *"templates"* for which arguments are replaced with multiple values and the answer is updated accordingly. For each template we automatically generate dozens of examples, and compare the accuracy of HYPOTHESIS-ONLY with the effect of adding the explicit knowledge. We evaluate the examples on the ALL COMBINED model. Results, shown in Table 2, display an average increase of 56% in accuracy when adding the explicit knowledge, achieving a high accuracy of 93.5% on average. This suggests that the model consistently combines multiple implicit skills, such as hypernymy, age comparison, year comparison, as well as explicit multi-hop reasoning.

| hypothesis + explicit knowledge template | template arguments | hypothesis only | with explicit knowledge |
|---|---|---|---|
| H: [NAME] can live up to [LIFE_SPAN] years. C: A Human can live to up an older age than a [ANIMAL_TYPE]. A whale can live up to 200 years. A [ANIMAL_TYPE] can live up to [ANIMAL_LIFE_SPAN] years. | [NAME]: John, Elizabeth, Chen, Richard, Don, Moses [LIFE_SPAN]: 5,10,70,80 ANIMAL_TYPE, LIFE_SPAN: (Horse:30), (Panda:20), (Cow:22), (Monkey:30), (Tiger:15) | 50.0 | 90.0 |
| H: [ENTITY] contains [PARTICLE]. C: A physical entity is made of matter. Matter contains [PARTICLE]. Energy does not contain [PARTICLE]. Abstract entities do not contain [PARTICLE]. | [PARTICLE]: Protons, Atoms, Neutrons, Quarks [ENTITY]: A Frog, A Car, A Mountain, A River, Light, Radiation, Electricity, Magnetism | 25.0 | 96.9 |

Table 2: An analysis of ALL COMBINED performance over a set of examples generated from templates. The template hypothesis and explicit knowledge are displayed on the left column. Template arguments, enclosed in brackets, with their respective values in the middle column. Hypothesis-only, and hypothesis with explicit knowledge accuracy, are shown on the right.