[Reviews · NeurIPS 2020]

Review 1

Summary and Contributions: The paper presents a systematic analysis of the ability of the RoBERTa LM to learn some rule-based reasoning skills, evaluated through a binary classification task over simple hypothesis (i.e., true/false). From my understanding, there are two main skills tested in the paper: (1) the ability to reason over hypernyms/meronyms and (2) the ability to count. Such abilities are assessed with varying amount of explicit knowledge provided as input to the LM. When oracle rules are provided (thogheter with some distractors) the model can exploit those, and provide almost always the correct answer. When only partial information is provided, the model still perform better than the context-agnostic version, suggesting that RoBERTa can exploit to some extent its implicit knowledge to fill the missing connections.

Strengths: - Assessing reasoning skills of transformers-based architectures is an hot research topic. - I liked the experiment presented in Section 4.2 "Analyzing systematicity". Is interesting to see that RoBERTa can reason over de-lexicalized entities, especially after the clever intervention to add explicit rules for implicit reasoning where the model fails. - I liked the experiment in Section 4.4, that highlights how reasoning skills can be combined by exposing the model to a mix of those during training.

Weaknesses: - synthetic data. I believe the main limitation of this work is the nature of the data: it relies on a combination of manually curated rules and symbolic representations of knowledge (e.g., Wikidata triples). It is unclear to me how such model will behave in a real-word setting, where human curated rules/triples are not available but natural language sentences were such rules/triples manifest. It should be easy to perform an experiment with natural sentences instead of triples, for instance, by exploiting resources such as https://github.com/uma-pi1/OPIEC or https://hadyelsahar.github.io/t-rex - the quality and usefulness of the distractors is unclear. In the supplementary files I can see that the performance of RoBERTa with or without those is close. In general, I would advice to better justify why distractors are needed in the first place. It will be interesting also to experiment with adversarial distractors (but that's orthogonal to the former point). - The authors argued that LM can be trained to reliably perform systematic reasoning. However, from Table 1 it seems that, given explicit knowledge the training of RoBERTa is irrelevant - it always get it right. It would be interesting to see if this behaviour can be observed with no further training, just relying on how RoBERTa was pre-trained.

Correctness: I struggle to see if the results of the experiments presented in Section 4.3 "Counting over Implicit Facts" are due to an artefact of the training/test data or there is something interesting there. First of all the exposition can be improved, the distinction (1, K-1) vs K is quite misleading and a better naming can help. As far as I understand when subset=K there are no other positive examples and it is unclear if the model predicts always *false* because it's counting or because it "knows" that the hypothesis are all false (given the context). A simple experiment that might help in this regards is to inject "false" facts in input that affects the counting. For instance, for the Beatles example, you can add "{Paul McCartney}/{George Harrison}/{Ringo Starr}/{Barack Obama} is member of The Beatles" (+ the fact that there are 4 members) and then ask "John Lennon is member of The Beatles". Will results be as high as 99.7% also in that case? Also, why in table 3 there is a subset associated with Hypothesis-only? That solution by definition doesn't have access to any context, or am I wrong?

Clarity: It can be improved in some parts. See the above section (Correctness) for some suggestions. Another aspect that was not easy to understand is the description of the meronyms zero-shot test set - an example could help to clarify.

Relation to Prior Work: A missing relevant work is "How Context Affects Language Models' Factual Predictions", Petroni et al., AKBC 2020.

Reproducibility: Yes

Additional Feedback:


Review 2

Summary and Contributions: This work attempts to analyze the ability of pre-trained models to *systematically* reason over a combination of explicit and implicit knowledge under a variety of settings. The observations are: 1. RoBERTa can reason over its implicit knowledge to classify a fact as true/false when given a context that gives the model some hints about the relevant implicit knowledge needed. 2. RoBERTa can also leverage its implicit knowledge to classify facts that require basic counting abilities. 3. Experiments show that models such as RoBERTa can combine different skills in a compositional way. The paper is well-written and considers a variety of experimental setups to carefully isolate confounding factors, however there are a few questions about the experiments that I elaborate on below.

Strengths: - The experimental setup / dataset creation is novel. - Section 4.2: Important questions around systematicity of reasoning skills are explored via a carefully designed experiment that adds a lot of depth to the analysis. - The analysis of looking at whether adding facts that the model doesn't know about increasing performance faster than adding facts that the model knows about, is also a very interesting experiment.

Weaknesses: Overall, I really liked the experiments in the paper, but some of the analysis could be made more complete. In particular, I had the following questions about the experiments: 1. In the experiments of Section 4.1, a possible explanation for the model's performance could be that it's not due to implicit reasoning but due to the distractor subject leaking the answer: For example, given A mammal has a belly button and A whale eats fish, the model can infer that a whale has a belly button simply by combining these facts instead of doing implicit reasoning. 2. What happens if you do not do "context" dropout at training time (removing certain parts of the context 50% of the times etc.)? 3. One possible explanation for why the hypothesis-only results are poor is because the hypothesis-only conditions are only seen in 20% of the examples. What happens if the model is re-trained with only hypothesis information and labels, and then evaluated in the hypothesis only mode? 4. To isolate the effect of pre-trained representations, why do the authors choose to use a different architecture (ESIM) instead of using RoBERTa with randomly initialized weights? 5. In the counting experiments, is the model really counting? A possible explanation is that the model doesn't really count but just looks at vector space similaries? For e.g. *Mick Jagger is a member of Beatles* is predicted as False, simply because it is far away in representation space from Ringo Starr and John Lennon, and not because the model counts. This could be tested by dropping the quantity fact and assessing performance. (To prevent train/test mismatch the model would need to be re-trained without the quantity fact).

Correctness: Yes

Clarity: Yes, the paper is overall well written with a lot of careful analysis. However, I believe some of the wording is a bit too strong: 1. The "combination of skills" aspect is only very weakly evaluated. Can the model compose more than 2 skills? It also seems like "sizes and age'year" are not seeing a very large positive transfer from COUNTING? 2. The presentation of results in section 4.4 seems rushed. What is the size of each of the sub-splits hypernyms/counting/sizes and age/year? The evaluation dataset seems very small, so reporting std dev. across several runs seems important.

Relation to Prior Work: Yes

Reproducibility: Yes

Additional Feedback: 1. I think this paper can benefit from explaining the usage of "distractors" better. A possibility would be to somehow add in the following thought experiment: a) If we do not have distractor sentences at all, the label of the property can be used to infer the label of the hypothesis and the model can just copy the label of the property *even* when the hypernym rules are removed. b) But if we add a distractor property, the model is forced to choose *which* of the properties it should condition its prediction on, and crucially, choosing the right property involves knowing the hypernym of the hypothesis subject. For example, to choose between "A dog has 4 legs" and "A crow does not have 4 legs" given the hypothesis "A cocker spaniel has 4 legs", the model can choose the right property (and hence get the right answer) only if it knows what the hypernym of cocker spaniel is. 2. I also think the presentation of experiment design in Section 4.2 can be improved.


Review 3

Summary and Contributions: This paper shows that pre-trained LMs can be trained to perform systematic reasoning combining both implicit pre-trained knowledge and explicit natural language statements, which paves a path towards open-domain systems that constantly improve by interacting with users.

Strengths: 1) Thorough experiments on both implicit and explicit information shows that extra background information is useful in question answering. With explicit performers slightly better than implicit, we can see that the pretrained language model has already learned some knowledge beforehand. 2) Further unbiased experiments are used to show scores improve significantly even when only part of the model’s implicit beliefs are correct. This shows that the performance in open-domain interactive systems could be greatly improved if the user could provide a fix-error statement if this experiment could be applied to more complicated multihop scenarios. This can be viewed as a form of “one-shot learning” that improves the model on-the-fly without further training

Weaknesses: Multihop is limited to 2hop only, further experiments are needed to apply the finding of this paper to multihop QA. Would this same finding applied to squad like QA dataset?

Correctness: Yes

Clarity: Yes. One minor issue is that in Figure 3, “Underlined hypernym rules are removed“. However, the last three lines are partially underlined but not hypernym. This may cause some confusion. Suggest to change partially underline to color label.

Relation to Prior Work: Yes, relevant discussions are made to textual entailment and multi-hop QA, with emphasis on extra “teaching” from user interaction, which is one key spotlight for the paper.

Reproducibility: Yes

Additional Feedback:


Review 4

Summary and Contributions: This paper proposes shows that it is possible to adapt pretrained language models (LMs) on-the-fly based on natural language text in order to correct the model's behavior. When an LM would answer a question incorrectly, the authors supplement the model with a hint or relevant piece of evidence in the form of natural language text and find that the model is then able to produce the correct answer. This results are a proof of concept that large, black-box LMs can be adapted/corrected in a natural way / potentially by non-expert users of the system, simply by providing relevant natural language text. -------------- Post Rebuttal: The rebuttal mainly addresses concerns from other reviewers, and the responses there seem reasonable to me. I should also clarify that the use of synthetic/simple task was only a minor negative point to me (compared to R1).

Strengths: See discussion above. In particular, it is nice to be able to adapt LMs on-the-fly with new information or alternative information in order to change its prediction, conditioned on this information (especially since the information is in natural language, which is the most natural/easy way for human users to provide feedback).

Weaknesses: The tasks in the paper are somewhat simple/synthetic, and the model itself requires training data to be able to use natural language instructions. Thus, from the results from this paper alone, it is not clear how general approach is. However, combined with recent, stronger LMs like GPT3 (which show similar/stronger adaptive ability), I think the approach will become quite viable in the future, without requiring any specialized training data or learning to incorporate additional natural language hints/corrections.

Correctness: I did not notice anything that was incorrect. I do think that it is somewhat misleading to describe the model as "learning" on-the-fly from the new information, as it is simply conditioning on new information rather than updating its parameters.

Clarity: The problem is well-motivated in the writing. I did find some of the task descriptions hard to follow, and the figures were somewhat helpful but not fully clarificatory for me, so I was still left a little unsure about the except setup/problem for some subsections.

Relation to Prior Work: The paper positions itself well in light of prior work, though I feel there could be a stronger connection to work in learning from natural language human feedback (which seems more relevant than some of the currently cited work). For example, there is a nice work in learning to correct image segmentations based on natural language feedback ("Guide Me: Interacting with Deep Networks") or correcting dialogue models based on natural language feedback during a conversation ("Learning from Dialogue after Deployment: Feed Yourself, Chatbot!")

Reproducibility: No

Additional Feedback:

[Author Response · NeurIPS 2020]

We thank the reviewers for their constructive feedback and for stating that our work answers important questions around systematicity of reasoning skills, and paves a path towards open-domain systems that constantly improve by interacting with users.

In this work we focus on testing weather models can systematically reason over implicit knowledge. Thus, some of the design choices such as the use of synthetic data, distractors, and certain training mixtures, were designed to give better control toward answering this novel research question.

Below we answer all questions and provide results for requested additional experiments.

**R1, R4:** *The use of synthetic data instead of natural language data.* (a) The experiment in Section 4.4 uses **natural (not synthetic) language**. We show that the model achieves high accuracy here after training on synthetic data. Moreover, As suggested by R4, recent work hints that LMs, such as GPT-3 do increasingly well in bridging the gap between synthetic and natural language. (b) Because our main research question is the systematicity of reasoning, we want a setup where we have full control over the data presented to the model. Synthetic data is necessary to explore such novel research questions. (c) We agree that testing on natural language is desired and for the final version we will paraphrase automatically generated data using crowdsourcing.

**R1:** *The quality and usefulness of the distractors is unclear.* Thank you for your comment. Distractors are *crucial* for training. Without distractors the models find biases in the data that hurt generalization. For example, in the counting experiments, if no distractor member facts are shown, the model learns to count sentences, and ignores their content, failing to pay attention to the actual subjects of the member facts. We will clarify this in the final version.

**R1:** *It would be interesting to see if this behaviour can be observed with no further training, just relying on how RoBERTa was pre-trained.* This experiment was performed in Figure 4 of the RuleTaker paper (Clark et al., 2020), showing that performance is poor with few training examples.

**R1:** *In the counting experiment, what happens if incorrect member facts are presented to the model when the subset=K* Thanks for this interesting suggestion. We conducted this experiment, and found that the model still predicts 100% false, regardless of if the member fact is correct, suggesting it is counting the relevant member facts, rather than knowing them in advance. We will add this experiment to the final version.

**R2:** *Can results in Section 4.1 be explained by distractor subject leaking?* As explained in line 172 "We create development and test sets... where the subjects and objects are disjoint from the ones in the training set". Because distractors are chosen from disjoint sets, leakage is not possible. We will clarify this in the final version.

**R2:** *What happens if you do not do "context" dropout at training time* Thank you for this question. Without dropout, the model learns to rely solely on explicit knowledge, achieving lower accuracy on implicit knowledge tests. The model predicts 'False' when relevant rules are missing from the explicit knowledge. We will add this to the camera-ready.

**R2:** *What happens if the model is re-trained with only hypothesis information and labels, and evaluated in hypothesis-only* Thank you for this suggestion. As suggested, we trained RoBERTa-Large on Hypothesis-only data and the Hypothesis-only test results are moderately higher: $65.2 \rightarrow 69.7$. This is still well below the Implicit-Reasoning accuracy . We will add this experiment to the camera-ready.

**R2:** *Why do the authors choose to use a different architecture (ESIM) instead of using RoBERTa with randomly initialized weights?* Because the size of the training data is relatively small, we were unable to train large transformer-based LMs directly on our data from scratch. Thus, we used the smaller ESIM + GloVe embeddings to compare to a model with less implicit knowledge.

**R2:** *In the counting experiments, is the model really counting? This could be tested by dropping the quantity fact.* This is an interesting suggestion. We conducted this experiment and accuracy drops from $73 \rightarrow 64.4$ (Table 3, counting $(1, K-1)$), which is similar to hypothesis-only $64.1$, suggesting that the model is using the quantity fact for counting.

**R3:** *Multihop is limited to 2hop only* We agree that this can be extended but chose to focus on combining implicit and explicit knowledge rather than on the inference chain length, as done in RuleTaker. As stated in our related work section (line 364-369), the focus on systematicity distinguishes us from works on multi-hop QA.

**R4:** *Synthetic data* Please see response to R1 above.

**All reviewers** *Clarifications, wording, figure 3, and missing reference* We will clarify the writing of the experiments, specifically section 4.3 as requested by, mend the wording, improve figure 3, and add the missing reference, as requested by the reviewers.

[Meta-Review · NeurIPS 2020]

All 4 reviewers support acceptance for the contribution. I believe the contribution is original and intriguing enough to merit a spotlight. This summary from R4 shows how the work in this paper opens new possibilities in NLP, complementing powerful adaptable models such as GPT-3. “This paper shows that it is possible to adapt pretrained language models (LMs) on-the-fly based on natural language text in order to correct the model's behavior. When an LM would answer a question incorrectly, the authors supplement the model with a hint or relevant piece of evidence in the form of natural language text and find that the model is then able to produce the correct answer. This results are a proof of concept that large, black-box LMs can be adapted/corrected in a natural way / potentially by non-expert users of the system, simply by providing relevant natural language text.” The following issues that have been pointed by several reviewers, though only as an initial cause for rejection by R1: - Distractor examples: the initial write-up is quite confusing, especially as they are not our usual counter-examples, and most reviewers, myself included, did not understand their role initially. The authors have clarified their explanation, and offered to update the final version. - Synthetic data: after discussion, most reviewers agree that it is acceptable.